



**Reinforce lake water balance components estimations by integrating water isotope**
**compositions with a hydrological model**
Nariman Mahmoodi[1, 2*], Ulrich Struck[3], Michael Schneider[1] Christoph Merz[1, 2]
[1] Department of Hydrogeology, Institute of Geological Sciences, Freie Universität Berlin, Berlin, Germany
[2] Lowland Hydrology and Water Management Group, Leibniz Centre for Agricultural Landscape Research
(ZALF), Müncheberg, Germany
[3] Museum für Naturkunde, Leibniz-Institut für Evolutions- und Biodiversitätsforschung, Berlin, Germany
[*]Corresponding author, email: n.mahmoodi@fu-berlin.de
**Abstract:**
Accurate estimation of water balance components of groundwater-fed lakes, including
subsurface inflow, as well as actual evaporation from lakes, poses a complex task for
hydrologists employing hydrological models. Hence, an alternative approach was used to
capture the dynamic behavior of the hydrological groundwater/surface water system, which
can be used for integration with the hydrological model and serves as a validation for the
hydrological model estimates of the water balance components. The approach, based on
measurements of the stable isotopes ($\delta^{18}$O and $\delta$D) enables the quantitative estimation of the
individual water flux and evapotranspiration rates. An isotope-mass-balance model was used
to quantify lake water balances over a one-year sampling period. The approach is based on the
global relationship between the $\delta^{18}$O and $\delta$D values of the precipitation and kinetic isotopic
fractionation in the lake water during evaporation. Assuming that the lake is hydrostatically
connected to the groundwater the isotope mass-balance model accounts for the quantification
of the evapotranspiration rate considering the groundwater inflow compensating the
evaporation loss. The study addresses the model-based quantification of subsurface-
groundwater inflow and evaporation losses of a young glacial groundwater lake (Lake Gross
Glienicke (GG), southwest of Berlin in the Havel catchment), over the period from 2015 to
2023 with the integrated hydrological model HydroGeoSphere. Utilizing the isotopic mass
balance model, HydroCalculator, under steady-state hydrologic regime conditions, the
evaporation-to-inflow (E/I) ratio is determined for the period of one year spanning August
2022 to September 2023. Employing the fully integrated hydrological model, calibrated and
validated under monthly normal transient flow conditions from 2008 to 2023 for the lake





catchment, subsurface, and groundwater inflows to the lake are calculated and compared to
the calculated E/I ratios based on the isotopic measurement of the lake water. Isotopic
signatures of surface water, groundwater, and rainwater ($\delta^{18}O$ and $\delta D$) confirm a flow-
through type for the lake. The calculated E/I ratio for GG Lake is around 40%. The calculated
evaporation for the years 2022 and 2023, within the isotopic mass balance model framework
($E_{iso22}$ = 601 mm, $E_{iso23}$ = 553 mm), aligns well with the actual evaporation from the lakes
calculated by the HGS model ($E_{HGS22}$ = 688 mm, $E_{HGS23}$ = 659 mm). The change in the ratio
of evaporation to inflow (E/I) leads to a significantly improved estimation of evaporation
rates after correction for temperature fluctuations and inflow data from previous years (2015-
2021). With a correlation coefficient of 0.81, these revised estimates show a high degree of
agreement with the evaporation rates predicted by the HydroGeoSphere (HGS) model for the
corresponding years. Despite the uncertainties associated with the analysis of the water
isotope signature, its integration into the hydrological model serves to validate the
hydrological model calculations of the water balance components.
**Keywords:** HydroGeoSphere, HydroCalculator, Lake Water Exchange, Evaporation Loses,
Stable Isotope, Gross Glienicke Lake

## 1. Introduction

Hydrological models have undergone substantial advancements in past decades (Singh, 2018;
Herrera et al., 2022), but still face unsolved problems and uncertainties in depicting
hydrological processes (Liu and Gupta, 2007; Renard et al., 2010). Recently, a wide range of
monitoring and modeling techniques have emerged for investigating water fluxes at different
scales (Fekete et al., 2006, Windhorst et al., 2014). However, limitations in the hydrological
model parametrizations lead to insufficient quantification of water flows, and therefore a
quantitatively and qualitatively incorrect interpretation of hydrodynamic processes and, as a
result, inaccurate assumptions for water management purposes (Müller Schmied et al., 2014).
Improving the informative value of the models, representing complex hydrological processes
is needed to enhance the applicability of the models for future estimations and scenario
analyses. The optimal adaptation of hydrological models to real conditions for a precise
determination of water balance components is often considered unattainable within the
current technical possibilities due to the overwhelming amount of work associated with field
measurements. Particularly the quantification of groundwater-surface water exchange using





the hydrological models faces pronounced uncertainties in considering geostructural
heterogeneities in different scales. To validate the model results, arduous monitoring surveys
are required to measure for example the groundwater-surface water interactions along rivers'
banks or lakes' shorelines (Partington, 2020). Hence, a combination of hydrogeological
modeling, field measurements, and innovative isotopic-based studies along with appropriate
linkage between these approaches will be a concrete way to achieve optimal parameterization
and validation capabilities for modeling hydrological processes in complex geohydraulic
systems.
Recent studies show that the stable water isotope mass balance on different water sources
together with the numerical models improve the model performance in simulating the
interactions between groundwater and surface water (e.g., Jafari et al., 2021). The isotopic
insights enhance hydrogeologists' efforts to calculate water balance components such as the
groundwater inflow to the surface water resources and water losses due to evaporation
(Skrzypek et al., 2015; Vyse et al., 2020). This method is based on the fractionation of heavier
isotopes caused by evaporation from surface water, provoking a disparity in isotopic
composition between groundwater and surface water. However, a limitation of this approach
is that the stable water isotope mass balance is restricted to the sampling time, making it
incapable of accounting for the transient behavior of groundwater-surface water exchange
over different time periods. Although water isotope tests can describe the changes in water
fluxes in specific time frame intervals (e.g. monthly), deficiency in providing spatial
exchanges between groundwater and surface water can be also introduced as their limitation.
These gaps can be addressed by integrating water isotope analyses with physical-based
hydrogeological models.
Water isotope analyses are helpful techniques to evaluate the model's performance. For
instance, Ala-Aho et al. (2015) assess the hydrogeological model performance by comparing
the simulated groundwater inflow to lakes in the middle of Finland with calculated recharge
by water isotope analyses.
In the northeast of Germany, groundwater levels and landscape runoff have largely been in
decline for over three decades (Lahmer 2003; Germer et al., 2011; Merz and Pekdeger, 2011);
regional climate studies suggest further decreases over the next decades (Gerstengarbe et al.,
2003, 2013). Thus, water resource management for this region requires a thorough assessment
of possible adaptions and measures to counteract or mitigate severe consequences, such as
decreasing groundwater heads and surface water levels and declining groundwater and surface



water quality. The development of integrated management schemes for groundwater-
dependent ecosystems such as lakes under climate changes, requires a more comprehensive
understanding of hydrological dynamics and a better estimation of ecologically relevant water
fluxes are of prime importance. Therefore, an alternative approach was developed to capture
the complex behavior of the hydrological groundwater-surface water system. This approach
can be integrated with hydrological models to improve parameterization and validate model
calculations in water balance components.

## 2. Materials and methods

### 2.1 Study area

Gross Glienicke Lake (GGS) with an area of 0.59 km$^2$ and a maximum depth of 10 meters is
located in Berlin-Brandenburg state, Germany (30–87 m.s.l., Fig. 1a). It is a young glacial
lowland lake that is exclusively fed by groundwater. The lake's water levels have shown
significant seasonal variability (around 0.4 m) over recent decades (Fig. 2). Since 2014 the
lake's water level has faced severe drops (Fig. 2) around 1.28 meters. GGS as a seepage lake
(lake without an outlet) is surrounded by low-density residences (Fig. 1). In the north and
northwestern areas, grassland and farmland are the dominant surface cover and directly overly
sandy soils. The aquifers are recharged in the Döberitzer-Heide region where fine sandy soil
with a hydraulic conductivity ranging from $8 \times 10^{-5}$ to $12\times 10^{-5}$ m/s can be found according to
the lab analysis. The root zone within the grassland ranges in depth from the surface to 30 cm.
The groundwater monitoring measurements illustrate a smooth hydraulic head gradient from
West to East, highlighting connections between the lake and the aquifer recharge area from
the Döberitzer-Heide region (Fig. 3). On the east side of the lake catchment, a regulated river,
the Havel, has been flowing from northeast to southwest. Continuous stratigraphic units have
been delineated throughout the lakes' catchment based on the geological features information
collected from 480 boreholes (Fig. 2, The Federal Institute for Geosciences and Natural
Resources - BGR 2022). The geology in the study area is formed by a series of layered
Pleistocene and Tertiary sediments that are approximately 150 to 200 m thick, with a lower
confining bed of Oligocene marine Rupel clay. The series consists of a complex interplay of
glacial deposits from the Pleistocene and permeable marine and limnic sediments of the
Upper Oligocene and Miocene. The series can be divided into an upper unconfined aquifer
system of shallow Weichselian and late Saalian sediments. In general, a shallow (i.e., 5 to 10





m) unconfined aquifer is separated from the thick (140 to 190 m) lower confined aquifer by a
15 to 20 m thick layer of Saalian sediments. The confined and unconfined aquifers consist of
multiple permeable sediment layers partially disconnected by layers of till, but still
hydraulically connected. The hydraulic connection to the lakes is mainly controlled by these
aquifer layers. Underneath these sediments is a thick confined aquifer system of the early
Saalian and Elster layers, and Upper Oligocene and Miocene sediments. The first shallow,
unconfined aquifer in the catchment area is characterized by highly permeable glacial sand
and gravel deposits (Holocene and Weichselian). A till layer (Weichselian age, Fig. 3) is
found in the underlying layers. The till is underlain by late Warthe sandy sediments forming
the second aquifer.

**2.2 Available meteorological data**
From 1990 to 2023, radar-based CER v2 data (The Central Europe Refined Analysis version
2, details on the data pre- and past-processing are provided by Jänicke et al., 2017) generated
a mean daily temperature of 10.4 °C and average annual precipitation of 612 mm, with an
average annual actual evapotranspiration (PET) of 639, 646, and 670 mm for farmland and
grassland, forest, and urban areas respectively. The annual mean humidity in Gatow station
varies from 50% to 70% over the last two decades (2000-2023). During the hydrological year
of the survey (August 2022 to September 2023), the mean precipitation, temperature, and
humidity in the study region were 51.9 mm, 12.3 ˚C (Potsdam weather station, DWD, 2024;),
and 69 % respectively (recorded in Gatow weather station of The Berlin Measurement
Network (MEVIS, Fig. 1). Precipitation data as one of the boundary conditions in the
modeling work has been obtained from the Potsdam station of the German Weather Service
(DWD, 2024).





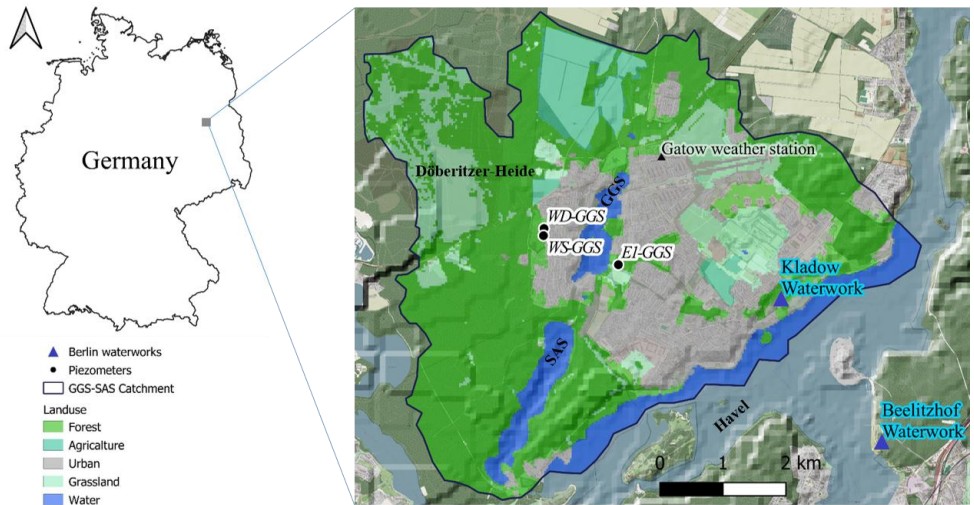


**Fig. 1.** Location of the study area, highlighting Gross Glienicke Lake (GGS), Sacrow Lake (SAS), Havel

channel, piezometers on the east (E-GGS) and west (WD-GGS and WS-GGS) side of GGS, Berlin waterworks,

and land use classifications. © OpenStreetMap contributors 2023. Distributed under the Open Data Commons

Open Database License (ODbL) v1.0..

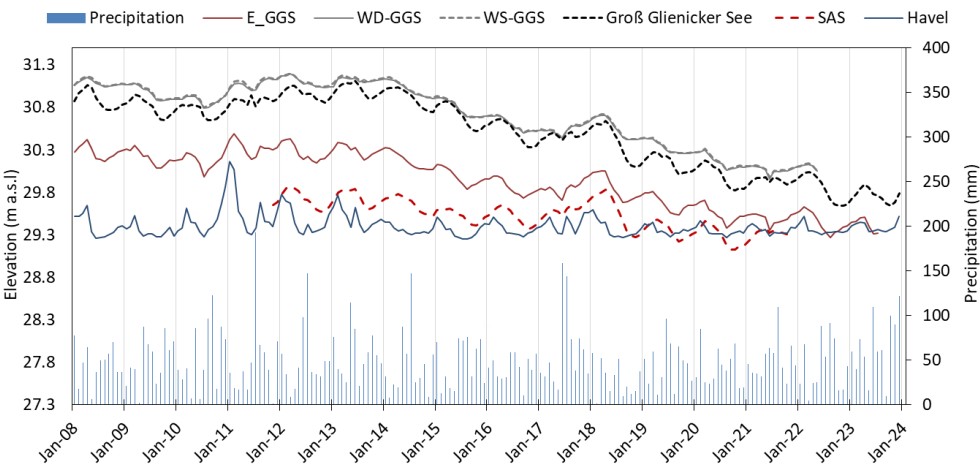


**Fig. 2.** Lakes water level fluctuations and precipitation variations during the period of 2008 to 2023.







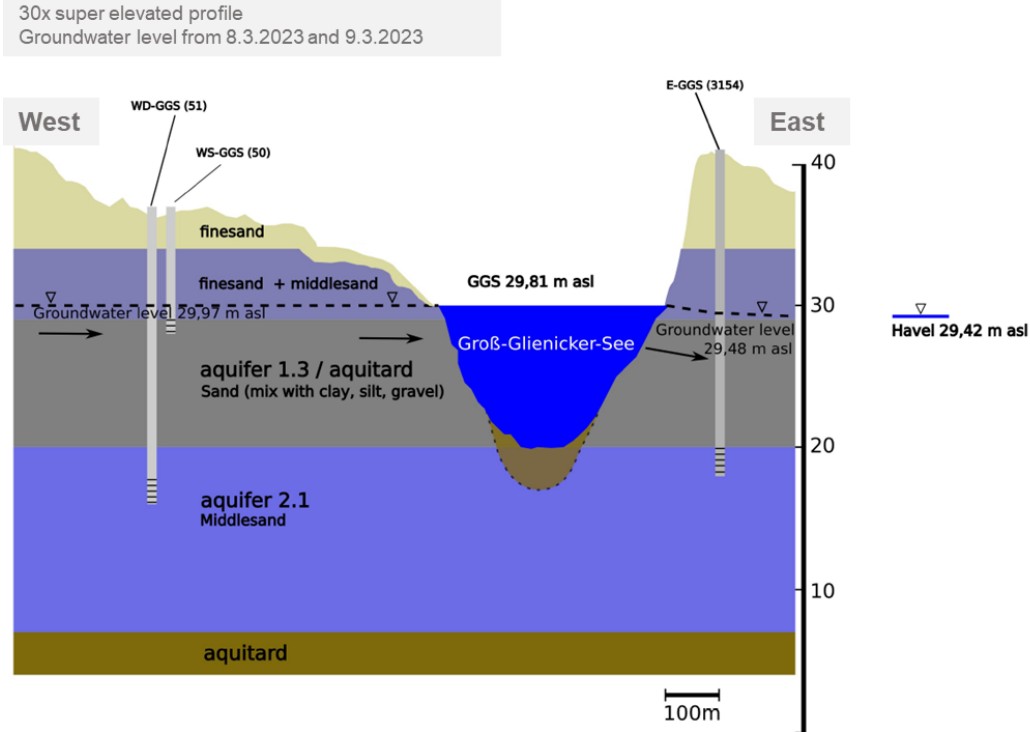


**Fig. 3.** Conceptual cross-section of the aquifer demonstrating the geological structure, the hydraulic head, and water flow direction from the groundwater recharge area towards Gross Glienicke Lake (GGS) and the groundwater discharge area with a lower altitude. WD-GGS, WS-GGS, and E-GGS are the piezometers on the west and east sides of GGS.

### 2.2 Stable isotope analysis

Surface water samples were collected for one year (from August 2022 to September 2023) on a monthly time interval from GGS and three piezometers installed in the first two aquifers, encompassing the lakes, and from two rainwater samplers. The total depth and well water volume of the monitoring well, and the stability of in-situ parameters such as temperature, pH, and electrical conductivity (EC) were monitored as guidance of appropriate timing for water sampling from the piezometers. All water samples collected in clusters within two-day excursions were filtered through a membrane filter (0.2 and 0.45-μm pore) and stored at 6 °C to prevent evaporation before laboratory analysis. Stable isotope ratios of oxygen ($^{18}O/^{16}O$) and hydrogen ($^{2}H/^{1}H$) in $H_2O$ in water samples were measured with a PICARRO L1102-i isotope analyzer. The L1102-i is based on the WS-CRDS (wavelength-scanned cavity ring-





down spectroscopy) technique (Gupta et al., 2009). Measurements were calibrated by the
application of linear regression of the analyses of IAEA calibration material VSMOW,
VSLAP, and GISP. The stable isotope ratios of oxygen and hydrogen are expressed in the
conventional delta notation ($\delta^{18}$O, $\delta$D) per mil (‰) versus VSMOW. For each sample 6
replicate injections were performed and arithmetic average and standard deviations (1 sigma)
were calculated. The reproducibility of replicate measurements is generally better than 0.1 ‰
for oxygen and 0.5 ‰ for hydrogen.

**2.3 Isotope mass balance model**

The evaporation loss from a lake such as GGS Lake can be calculated by knowing the
transient stable water isotope compositions of inflow and moisture in ambient air and climate
data (air temperature and humidity) for the specific period of time and considering a steady-
state hydrologic condition (no additional water inflows, Tweed et al. 2009; Gibson and Reid,
2010). The isotope mass balance model (Hydrocalculator) whose capability has been verified
through various field experiments globally (Skrzypek et al. 2015; Vyse et al., 2020) was
applied to estimate evaporation over inflow ratio (E/I) for the GGS Lake in the steady state
condition. The differences between stable isotope compositions of water samples reflect the
isotopic phases: enrichment (heavier isotope) or dilution (lighter isotope). Hence, a series of
time-based analyses enables the assessment of evaporation progress. Climate data from
nearby weather stations (Gatow and Potsdam) were utilized to address uncertainties arising
from the distance to the points of water samplings (Gibson and Reid, 2014; Skrzypek et al.
2015). The stable isotope composition of moisture in ambient air ($\delta_{air}$) is estimated from the
mean monthly weighted averages from the stable isotope composition of precipitation ($\delta_{pcp}$)
of GNIP station (GNIP/Berlin (DWD, BFG, BGR & HHZM, Stumpp et al. 2014 and Schmidt
et al. 2020) which were corrected by local precipitation stable isotope composition. The $\delta_{air}$ is
calculated based on the rain and rain-LEL as follows (Gibson and Reid, 2014):
$\delta_{air} = (\delta_{pcp} - \varepsilon^{+})/(1 + \varepsilon^{+} \times 10^{-3})$                                        **Eq.**
**1**
where $\varepsilon^{+}$ is an isotope fractionation factor that is solely temperature-dependent. $\varepsilon$ is the total
fractionation factor, and equals the sum of the equilibrium isotope fractionation factor $\varepsilon^{+}$, as
given above plus the kinetic isotope fractionation factor $\varepsilon_{k}$ (Gibson and Reid, 2010):





$\varepsilon = \varepsilon^+ / (1 + \varepsilon + \times 10^{-3}) + \varepsilon_k$                    **Eq.**
**2**
The kinetic fractionation $\varepsilon_k$ is defined as (Gat 1995):
$\varepsilon_k = (1 - h) \times C_k$                    **Eq.**
**3**
According to Gonfiantini, 1986 and Araguas-Araguas et al., 2000, the kinetic fractionation
constant ($C_k$) is 12.5 percent for $\delta D$ and 14.2 percent for $\delta^{18}O$. Air relative humidity (h) is
given as a fraction.
Based on the local climate conditions the enrichment of stable isotope compositions can be
limited. According to Gat and Levy (1978) and Gat, (1981), this limitation threshold ($\delta^*$) can
be estimated by considering air humidity (h), $\delta_{air}$, and a total enrichment factor ($\varepsilon$).
$\delta^* = (h \times \delta_{air +} \varepsilon) / (h - \varepsilon \times 10^{-3})$                    **Eq.**
**4**
When this limitation exceeds, further evaporation does not result in isotope enrichment.
The ratio of evaporation over inflow (E/I) can be calculated using the following reformulated
equation (e.g. as by Mayr et al. (2007)) under steady-state hydrological conditions. E/I is the
fraction of inflowing water evaporated from GGS Lake:
$E/I = ((\delta_{inflow} - \delta_{outflow}) / (\delta^* - \delta_{inflow}) \times E_s)$                    **Eq. 5**
enrichment slope ($E_s$) is defined by Welhan and Fritz, 1977 and Allison and Leaney, 1982
accordingly:
$E_s = (h - (\varepsilon \times 10^{-3})) / (1 - h + (\varepsilon \times 10^{-3}))$                    **Eq.**
**6**
The model calculates the evaporative losses based on the theory behind the Craig–Gordon
model (Gibson and Reid (2014)). The variables used in the Hydrocalculator model are listed
in Table 1.
**Table 1.** The list of variables used in the Hydrocalculator model

| Variable | Description | Unit |
|---|---|---|
| $T$ | temperature | ˚C |
| $h$ | air relative humidity | - |





| $\delta_{air}$ | stable isotope composition of moisture in ambient air | % |
|---|---|---|
| $\delta_{pcp}$ | stable isotope composition of precipitation | % |
| LEL | slope of the local evaporation line | |
| $\varepsilon$ | total isotope fractionation | % |
| $\varepsilon^+$ | equilibrium isotope fractionation factor | % |
| $\varepsilon_k$ | kinetic isotope fractionation factor | % |
| $C_k$ | kinetic fractionation constant | |
| $\delta^*$ | limiting isotopic composition | % |
| E/I | Evaporation over inflow ratio | % |
| $\delta_{inflow}$ | stable isotope composition of inflow (groundwater) | % |
| $\delta_{outflow}$ | stable isotope composition of outflow (lake) | % |
| $E_s$ | enrichment slope | - |


### 2.4 Model domain configuration and boundary conditions


#### 2.4.1 Surface – subsurface flows


The HydroGeoSphere (HGS) modeling code (Aquanty Inc, 2023) was used to simulate the
hydrological processes in the GGS Lake catchment. HGS is a 3-D, fully integrated, and
physically-based model with the capacity to simulate the interwoven flow mechanisms of
subsurface and surface water by coupling solutions obtained from the diffusion-wave of the
two-dimensional, depth-integrated diffusion-wave of the Saint Venant equation governing
surface water flow (Eq. 8, Viessman Jr. and Lewis, 1996) and the Richards' equation
governing three-dimensional unsaturated and saturated subsurface flows (Eq. 9).

246                                                                                                 *Eq. 7*

$$\frac{\partial \phi_0 h_0}{\partial t} - \frac{\partial}{\partial x}\left(d_0 K_{0x} \frac{\partial h_0}{\partial x}\right) - \frac{\partial}{\partial y}\left(d_0 K_{0y} \frac{\partial h_0}{\partial y}\right) + d_0 \Gamma_0 \pm Q_0 = 0$$

$\phi_0$ represents the porosity (dimensionless) of the surface flow domain, which varies based on
the presence of rills and obstructions. $h_0$ stands for the water surface elevation (L). $t$ denotes
time (T). $d_0$ indicates the depth of flow (L). $K_{0x}$ and $K_{0y}$ represent surface conductance. $\Gamma_0$ is
the water exchange rate ($L^3 L^{-3} T^{-1}$) occurring between the surface and subsurface systems.
$Q_0$ represents external sources or sinks.
The interaction between the two flow domains is facilitated by the exchange term $\Gamma_0$ through:

254                                                                                                 *Eq. 8*





$$d_0\,\Gamma_0 = \frac{k_r K_{zz}}{l_{exch}}\,(h - h_0)$$
$k_r$ symbolizes the exchange's relative permeability. $K_{zz}$ represents the saturated hydraulic
conductivity in the vertical direction. $l_{exch}$ corresponds to the coupling length.
*Eq. 9*
$$\nabla \cdot (W_m\,q) + \sum \Gamma_{ex} \pm Q = W_m\left(\frac{\partial}{\partial_t}\right)(\theta_s S_w)$$
In the given context: $Wm$ (dimensionless) represents the volumetric porosity fraction within
the porous media domain. $\Gamma_{ex}$ stands for the volumetric exchange rate ($L^3\,L^{-3}\,T^{-1}$) occurring
between the porous media and other flow domains. $Q$ denotes the source or sink term. $t$
signifies time (T). $\theta_s$ corresponds to porosity (dimensionless). $S_w$ refers to the degree of water
saturation (dimensionless).
The flow rate $q$ ($L\,T^{-1}$) is portrayed as:
*Eq. 10*
$$q = -K \cdot k_r\,\nabla h$$
$K$ signifies the hydraulic conductivity ($L\,T^{-1}$). $k_r$ represents the relative permeability
(dimensionless), which is dependent on water saturation. $h$ corresponds to the hydraulic head
(L), calculated as the sum of the elevation head and pressure head.
The three-dimensional surface-subsurface flows in porous media and saturated zones were
solved with the control volume finite element method. Nonlinear equations were linearized
using Newton-Raphson and solved iteratively at each time step for the entire hydrologic
system.

### 2.4.2 Evapotranspiration

The evapotranspiration process needs specific prerequisites for accurate parameterization as it
is treated to play a dual role in the HGS as both a boundary condition and a distinct domain.
Within this framework, the evapotranspiration fluxes encounter a restriction governed by a
potential evapotranspiration flux (PET) which is defined by the modeler. The PET values are
designated a boundary condition, serving its purpose on the surface domain. With each
subsequent time step, a condition emerges, if the calculated actual evapotranspiration (AET)





surpasses the PET, then the PET value is employed as a flux directed toward the relevant
model faces. Conversely, if the calculated AET falls short of the PET, the computed AET
value itself becomes the applied flux. Additional details on the evapotranspiration process
formulations within the HGS model are presented in Kristensen and Jensen (Kristensen and
Jensen, 1975). The two-dimensional PET database used in this research is calculated using the
energy balance method and covers the period 2000 - 2022. The method is a balance of the
energy terms which are the net radiation, the change in the heat content of the lake, and the
latent and sensible heat fluxes. The equation is based on measurements of global radiation, air
and water temperature, cloud cover, and vapor pressure. The latent heat flux, which represents
the energy used for evaporation, was determined by subtracting the sensible heat fluxes and
the change in the heat content of the lake from the net radiation. The evaporation rate was
then calculated by dividing the latent heat flux by the latent heat of vaporization of the water.
A maximum evaporation threshold of 15 mm/day was set. More details are given by Ölmez et
al., 2024.
The simulation domain encompasses the entire GGS Lake catchment (Fig. 4) which is defined
based on the surface topography considering the equipotential lines derived from the lakes'
levels and measurements of the hydraulic head surrounding the lakes (piezometers). The
surface topography across the catchment was produced by stitching a digital elevation model
(DEM) from The Shuttle Radar Topography Mission (SRTM) with a resolution of 30 m, and
the bathymetry data of GGS (Wolter, 2010) and Sacrow Lake (SAS) (Lüder et al., 2006,
Bluszcz et al., 2008). Due to the high vegetation density, flat elevations, and the substantial
hydraulic conductivity of the predominantly sandy soil, the absence of river formation is
currently observed in the catchment. The foundational 2-D triangular mesh supporting the
comprehensive 3-D triangular prism grid within the HGS model was created using AlgoMesh
(Merrick, 2017). Each 2-D mesh layer encompasses a total of 2837 mesh nodes and 5300
triangular finite elements (Fig. 4). The complete 3-D model (Fig. 4) grid extends the 2-D
mesh across 15 subsurface layers, broadly categorized as one soil layer, 14 Quaternary
material layers, and one competent bedrock layer (Rupel clay).





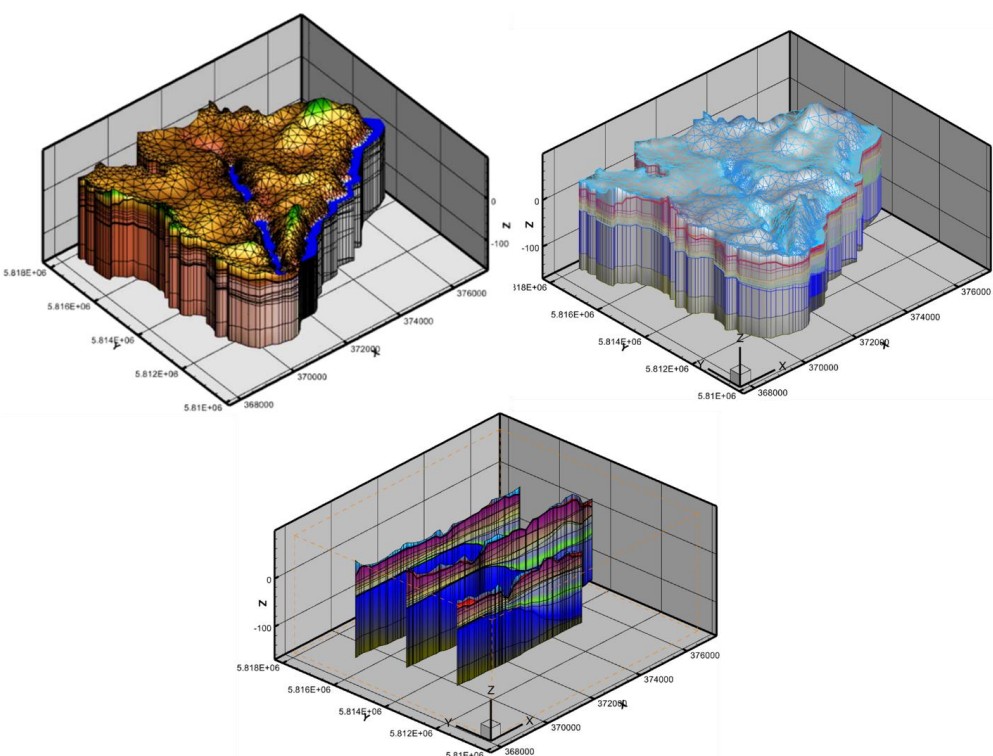


**Fig. 4.** Hydrostratigraphic units and enlarged view of the mesh within the Gross Glienicke

Lake catchment

Spatially distributed land cover data (Fig. 1) were utilized to capture a broad range of factors
influencing evapotranspiration and overland flow, including evaporation depth, root depth,
leaf area index (LAI), surface roughness, rill storage height, and obstruction storage height.
Specific parameters for evapotranspiration (ET) and overland flow are tailored to each land
cover type. To accurately reflect the impact of vegetation growth on water demand through
evapotranspiration, the Leaf Area Index (LAI) during winter (January) and summer (July)
using the Sun Sacan device type SS1 were measured, capturing both maximum and minimum
current LAI values. The measured LAI indices were then compared with data from the
MCD15A2H Version 61 Moderate Resolution Imaging Spectroradiometer (MODIS), which
provides a 4-day composite with a pixel size of 500 m for January and July (Myneni et al.,
2021). Corrected monthly average MODIS LAI values for each land cover type, spanning
from 2000 to 2023, were subsequently integrated into the HGS model.
**2.4.3 Unsaturated zone**





The top subsurface layer in the 3-D mesh with spatial varying depths shows the distribution of
soil materials across the catchment. The soil data were obtained from the soil map with a
scale of 1:200,000 (BUEK200) which was prepared by the Federal Institute for Geosciences
and Natural Resources (BGR, 2007). The soil samples were collected from various depths,
extending up to 3 meters, at 10 different sites, primarily in the groundwater recharge area
(Döberitzer-Heide region) and natural conservation zones. The sampling locations were
selected based on soil types. The percentages of sand, silt, and clay for each soil type were
determined in the laboratory to classify the soil textural types, using the United States
Department of Agriculture (USDA) soil textural calculator. A set of 2 soil textural types,
sand, and loamy sand has been recognized. Unsaturated soil hydraulic parameters and soil
moisture retention properties required for the van Genuchten application with the HGS model
were uniquely estimated for each soil type using the ROSETTA program, version 1 (Schaap
et al., 2001).
Underlying the soil layers are 14 Quaternary geology layers that overlie bedrock. To represent
the topography of the subsurface in the lake catchment, relevant data was extracted from the
groundwater model provided by Berliner Wasserbetriebe. This model (software FEFLOW)
was calibrated in 2012 and updated in 2013 (BWB, 2012 and 2013) using measured data from
the year 2010. The model focuses on analyses of the waterworks at Beelitzhof, Tiefwerder,
and Kladow. It, therefore, covers areas along both banks of the river Havel. Additional
datasets from boreholes were merged into a single surficial geology dataset using the
Rockware model setup for the GGS Lake catchment (Hermanns, 2022). The initial hydraulic
conductivity values for each type of Quaternary material were taken from the FEFLOW
model. The hydraulic properties of the unsaturated zone were manually adjusted during
manual model calibration.
**2.4.4 Groundwater – lake levels loggers**
A total of 8 groundwater monitoring wells scattered within the catchments, along with two
loggers set on the lakes, provide a well spatially distributed dataset of groundwater-lake level
dynamics for evaluating model simulations. The groundwater monitoring wells were selected
based on location, catchment area, and data availability spanning from 2000 to 2023. GGS
has been monitored since 1964. Moreover, within the study catchment, the regulated flow
system of the river Havel is maintained to facilitate water conveyance. Since 1980, an
established logger has been operational to meticulously monitor the water level dynamics
within this river. For the presented study particular interest lies in loggers No. 51, 52 (WD-


GGS and WS-GGS), and 3154 (E-GGS). WD-GGS(50) and WS-GGS(51) belong to two
different aquifers (shallow(WS-GGS) and deep aquifer (WD-GGS) and are situated on the
western side of GGS (Brandenburg, Fig. 3). These loggers in the recharge area of the GGS
lake consistently maintain water levels averaging 15-20 cm higher than the GGS.
Additionally, logger E-GGS, located close to the eastern shoreline downstream of GGS
(Berlin), consistently registers water levels averaging 30-40 cm lower than the GGS lake (Fig.

366    3).

### 367    2.4.5   Groundwater abstractions

Two major drinking water supply systems, Kladow and Beelitzhof, located alongside Havel
on the southwest side of Berlin have been in operation since 1888 and 1932 respectively by
BWB. Kladow comprises 16 wells up to 93 meters deep and a maximum pumping rate of
30,000 $m^3$/day, while Beelitzhof has 85 wells up to 170 meters deep and a maximum pumping
rate of 160,000 $m^3$/day. To assess the impact of groundwater withdrawals from deeper layers,
the model domain was extended to a depth of 150 meters below sea level. According to
studies BWB, upto 80 percent of the water extracted by Kladow originates from bank
filtration along the river Havel. The remaining 20 percent of the extracted water originates
directly from groundwater recharge as well as outflow from GGS. As various water resources
contribute to the overall drinking water production in the main waterworks in this area
(Beelitzhof, located on the western side of the Havel), a detailed analysis was conducted to
assess the share of the GGS Lake catchment. The analysis involved the implementation of
distinct scenarios within the hydrologic model.

### 381    2.4.6   Model evaluations

This study emphasizes the importance of a multifaceted approach to evaluate hydrological
model performance, utilizing both traditional and innovative methodologies. Initially to
evaluate model performance the simulated seasonal and long-term groundwater and lake level
fluctuations will be compared to observed water levels of the lakes and piezometers around
the lakes. The performance evaluation of hydrological models commonly relies on various
metrics such as the Nash-Sutcliffe efficiency (NSE), percent bias (PBIAS), root mean squared
error (RMSE), and the Kling-Gupta efficiency (KGE). The KGE, introduced by Gupta et al.
(2009), offers a comprehensive assessment by considering bias, correlation, and variability
separately. Given the specific hydrological focus of each metric, a multi-metric approach was
adopted for calibrating the HGS model parameters, as demonstrated to efficiently balance





model performance by previous studies (Pfannerstill et al., 2014; Mahmoodi et al., 2020). For
model assessment, NSE, PBIAS, RMSE, and KGE were employed as performance metrics on
a monthly basis. Calibration runs were evaluated based on predefined thresholds for NSE
(0.65), PBIAS (-25% to 25%), and KGE (0.65) to identify the most suitable configurations.
The calibration process was carried out manually due to the model's long execution time and
limited computational capacity. Emphasis was placed on calibrating model parameters of the
unsaturated zone which governs water movement into the soil and subsequently into or out of
the aquifer. The initial hydraulic conductivity values for each type of soil were determined
from existing literature (Steidl et al 2023) and lab analysis and later manually adjusted during
model calibration. The model parameterization for the saturated zone was initially derived
from the FEFLOW model calibrated by the BWB (BWB, 2012/2013). The calibration and
validation periods chosen for the simulation runs were 2008-2018 and 2019–2023,
respectively, preceded by an eight-year spin-up phase before 2008 to reach quasi-steady state
condition fitting to the conditions in 2008.
To evaluate the performance of the HydroGeoSphere (HGS) model on different angles, a
detailed assessment involving the simulation of the inflow to GGS Lake (denoted as $I_{HGS}$) was
undertaken. This parameter ($I_{HGS}$) was subsequently used as a testing parameter to evaluate
the model's performance in calculating evaporation rates for the years 2022 and 2023 using an
independently determined E/I ratio. The evaporation rate ($E_{ISO}$) can be expressed as:

411                                                                                  ***Eq. 11***

$$E_{ISO} = \frac{1}{A} \cdot I_{HGS} \cdot P$$
*Where A* is the area of the lake water body (m$^2$), *$I_{HGS}$* represents the annual inflow to the lake
(m$^3$) and *P* is the percentage of losses of inflow due to evaporation derived from the isotope
analysis (*E/I*). Figure 5 shows the methodology used to evaluate the model performance
across different dimensions. The underlying assumption for this evaluation is that an accurate
simulation of inflow to the lake by the HGS model ($I_{HGS}$) would yield evaporation rates ($E_{ISO}$)
comparable to those calculated by the HGS model ($E_{HGS}$). Thus, the consistency between
evaporation estimates derived from both approaches serves as a validation of the HGS
model's capability to simulate other water balance components precisely.




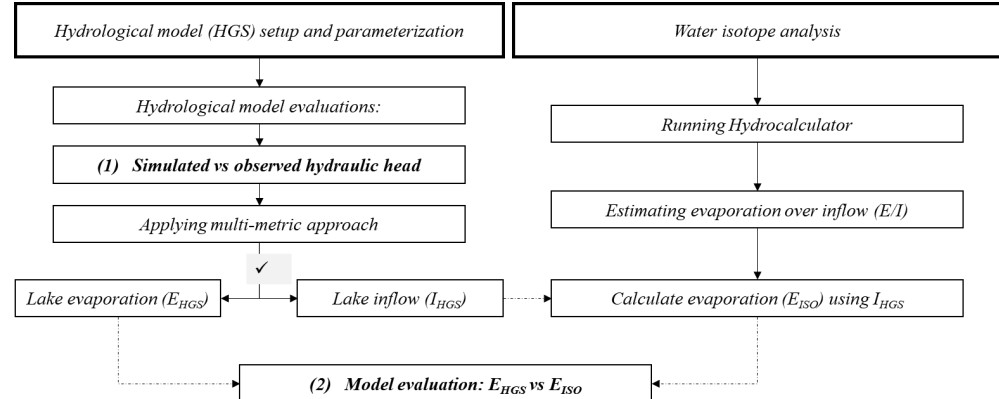


**Fig. 5.** Flow chart of the methodology employed to evaluate the model performance

The well-captured inflow and subsequent evaporation rates for the years 2022 and 2023 by
the HGS model, allow us to extend this approach for estimating evaporations during the
earlier period from 2015 to 2021. This period is crucial as it encompasses years without water
isotope analyses, during which significant drops in lake and groundwater levels were
observed. However, the E/I ratio derived from recent years (2022 and 2023) cannot be
directly applied to earlier years due to variations in temperature and inflow, which are key
factors influencing isotopic signatures (dilution and enrichment). To adjust the E/I ratio for
earlier years, we incorporated annual temperature variations and inflow data into our model.
Specifically, we compared the temperature and inflow of each specific year ($_{Yx}$) to the
corresponding values from 2022 and 2023($_{Y22-23}$). This comparison yielded ratios for
temperature ($T_{Yx}/T_{Y22-23}$) and inflow ($I_{Y2023}/I_{Yx}$), which were used to modify the E/I ratio
accordingly. For instance, to apply the $E/I_{Y22-23}$ to the year 2015, we multiplied the $E/I_{Y22-23}$
ratio by the temperature ratio ($T_{Y15}/T_{Y22-23}$) and the inflow ratio ($I_{Y2023}/I_{Y15}$). A temperature
ratio greater than 1 indicates higher temperatures in 2015 compared to 2023, suggesting a
higher E/I ratio, greater evaporation, and enrichment. An inflow ratio greater than 1,
indicating lower inflow in 2015 compared to 2023, would lead to a greater E/I ratio, reflecting
greater evaporation and enrichment. The adjusted E/I ratios were then applied to refine the
initial evaporation estimates from the isotopic mass balance model. These revised evaporation
estimates were subsequently compared to the evaporation rates calculated by the HGS model
for the period 2015-2021.

**3. Results**





**Isotopic analysis**

Alterations in the mean monthly isotopic compositions of lake water and groundwater, along with temperature and precipitation data, from August 2022 to August 2023 is presented in Figure 6. The δD values for GGS (Fig. 6a), show significant variability, showing a pronounced drop in δD values from around -8‰ in August 2022 to -17‰ in January 2023, followed by a gradual decrease (except February) to approximately -16‰ by June 2023, which represent a strong dilution phase. The period from July to September demonstrates enriched values of δD alongside rising temperature and evaporation as a consequence. The $\delta^{18}$O values for GGS (Fig. 6b) record a fluctuating pattern, ranging from -0.27‰ to 1.4‰, with peaks observed in August 2022 and July 2023, but experienced noticeable drops in April and May 2023. Overall, the isotopic data indicate that GGS experiences great isotopic enrichment (heavier isotopic composition).

The $\delta^2$H values of groundwater on the east side of GGS (E-GGS, Fig. 6c) range from approximately -40‰ to -50‰, with notable fluctuations throughout the year. The isotopic composition of groundwater on the west side of GGS (W-GGS, Fig. 6c) has less variability, with δD values mostly remaining between -55‰ and -60‰, suggesting a rather stable isotopic environment. Despite the fluctuations in δD values, the $\delta^{18}$O values (Fig. 6d) show less variation, indicating some degree of isotopic stability in the oxygen isotopes in the groundwater of both sides of lakes. E-GGS presents a relatively stable trend with $\delta^{18}$O values fluctuating between -5‰ and -6‰. W-GGS, with a seasonal pattern similar to E-GGS, shows a consistent range of $\delta^{18}$O values between -8‰ and -8.5‰ with minimal fluctuations (Fig. 6d). Overall, the E-GGS with heavier isotopic signatures experiences greater isotopic variability, meanwhile, the W-GGS site maintains a more consistent isotopic signature, indicative of a more stable hydrological regime. These observations (Fig. 6a,b,c,d) indicate that the isotopic composition of both lake water and groundwater was generally heavier (stronger enrichments) during the summer of 2022 compared to the summer of 2023.

The monthly average temperature (TMP, Fig. 6e), follows a clear seasonal pattern. It drops from around 20°C in August 2022 to a low of 5°C in January 2023, then rises again to about 20°C by July 2023. Alongside temperature, precipitation values fluctuate significantly, with peaks exceeding 90 mm in August 2022 and June 2023, and lower values around 20 mm observed in Oct and November 2022 and May and Septemeber 2023.

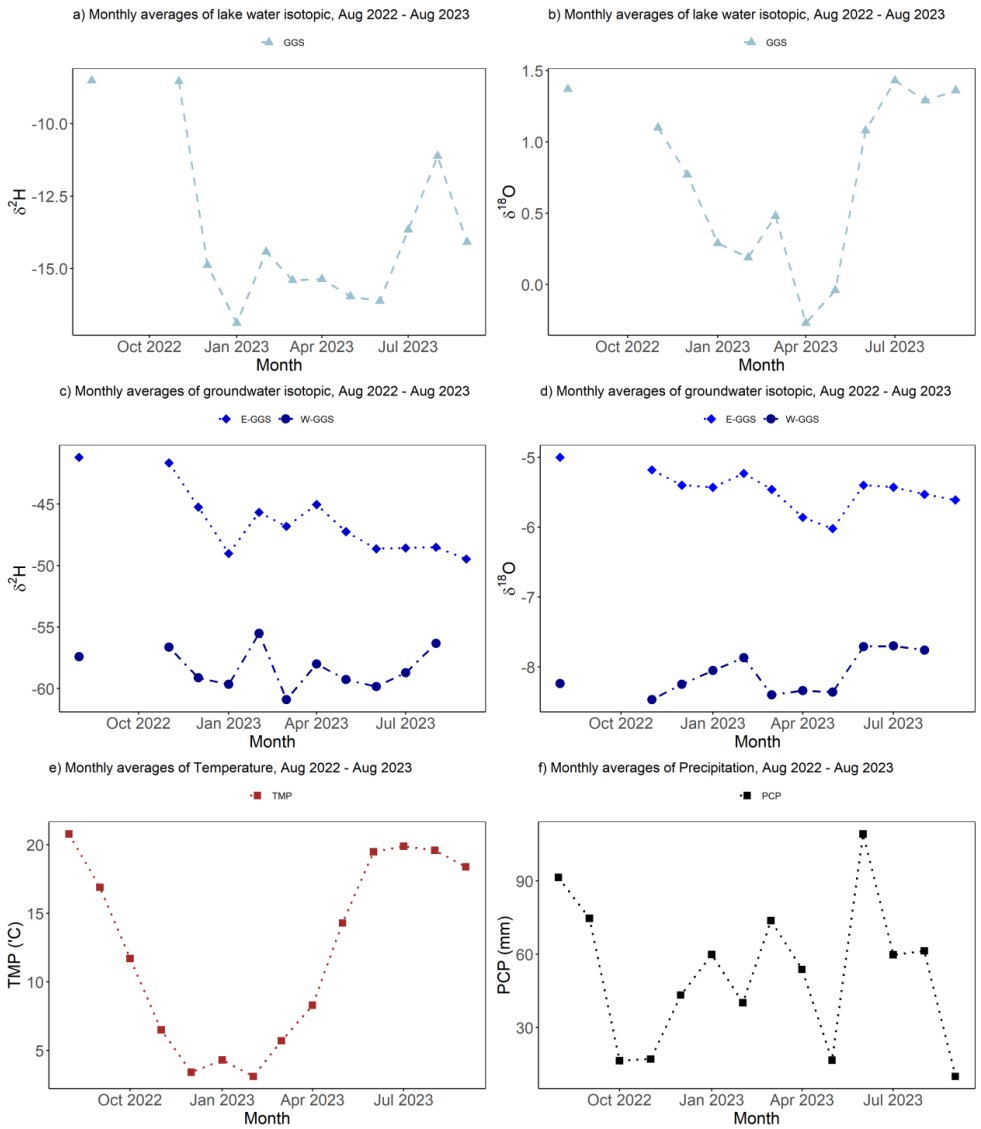

**Fig. 6.** Monthly averages of lake water isotopic compositions (a: δD and b: δ¹⁸O), groundwater isotopic compositions (c: δD and d: δ¹⁸O), temperature (e), and precipitation (f) data from August 2022 to September 2023.

The relationship between δD and δ¹⁸O values for lake water (GGS) and groundwater (WGGS and EGGS) from August 2022 to September 2023 are illustrated in Figure 7. The isotopic values for lake water are significantly clustered, with δD values between -25‰ and -5‰ and δ¹⁸O values from 2‰ to -2‰ and are isotopically heavier compared to precipitation and groundwater, suggesting significant evaporative enrichment. WGGS exhibits δ²H values from





-50‰ to -65‰ and $\delta^{18}O$ values from -9‰ to -7‰. EGGS $\delta^2H$ values range from
approximately -50‰ to -40‰, with $\delta^{18}O$ values between -7‰ and -5‰, indicating less
enrichment compared to lake water and higher enrichment compared to the groundwater on
the west side.

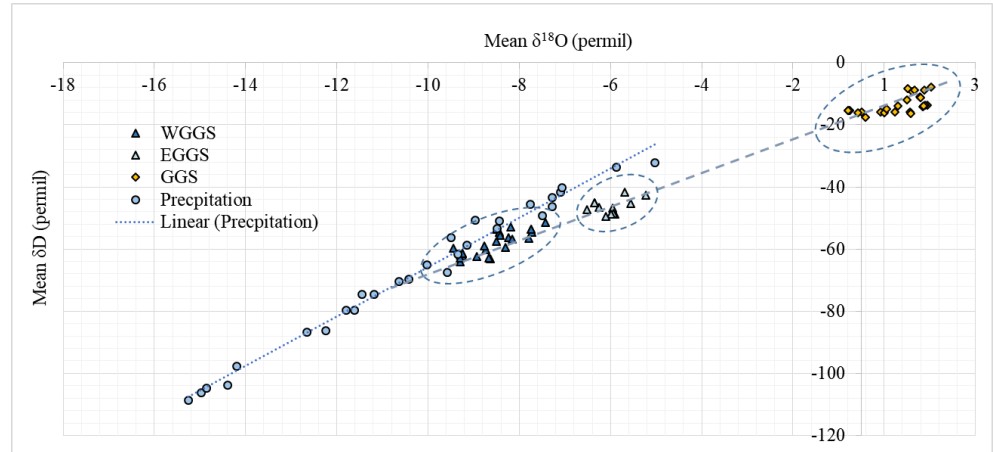


**Fig. 7.** Isotopic composition of lake water and groundwater measured on the west (WGGS) and east (EGGS)
sides of Groß Glienicke from August 2022 to September 2023. The local meteoric water line (LMWL) is driven
from (GNIP/Berlin (DWD, BFG, BGR & HHZM, Stumpp et al. 2014).

Variables used for the calculation of evaporative losses and evaporation over inflow ratio
(E/I) ratios calculated for GGS during the August 2022–September 2023 period are presented
in Table 2. The winterwater isotopic compositions (dilution phase) served as the initial
sampling point for calculating the E/I ratio in both years 2022 and 2023. The $\delta_A$-value of the
ambient air moisture was calculated based on the stable isotope composition of local
precipitation sampled in the Groß Glienicke region and the Lankwitz campus of the Freie
University Berlin. The calculated evaporative losses over inflow were equal to 43.4% and
42.3% based on $\delta D$ and 30.11% and 29.4% based on $\delta^{18}O$ in 2022 and 2023 respectively. The
E/I ratio calculated based on $\delta D$ is around 12% higher compared to the E/I based on $\delta^{18}O$.
Therefore, as a mean ratio, an average of 37% will be used for furhter analyses.

**Table 2.** Variables used for calculation of evaporative losses and the ratio of total evaporation to inflow (E/I) as
a function of the measured $\delta D$ and $\delta^{18}O$ isotope enrichments for Lake Gross Glienicke  (GGS) surveyed during
the August 2022–September 2023.





| Parameters | Description | $\delta D$ | $\delta^{18}O$ |
|---|---|---|---|
| $\varepsilon$ | Kinetic isotope fractionation factor [‰] (h dependent) | -887.5 | -1008.2 |
| $\varepsilon^*$ | Equilibrium isotope fractionation factor [‰] (T dependent) | 78.7465 | 9.3468 |
| $\varepsilon$ | Total isotope fractionation [‰] | -814.5018 | -998.9398 |
| Ck | Kinetic isotope fractionation constant [‰] | 12.5 | 14.2 |
| $\alpha^*$ | Equilibrium isotope fractionation factor [‰] (T dependent) | 1.0787 | 1.0093 |
| $\delta^*$ | Limiting isotope composition | -134.7724 | -30.2151 |
| m | Enrichment slope | -1.0129 | -1.0138 |
| $\delta_A$ | Ambient air moisture | -124.9844 | -16.7601 |
| $E/I_{Y2022}$ | Evaporation over inflow ratio [‰] of Groß Glienicke Lake in 2022 | 43.37 | 29.63 |
| $E/I_{Y2023}$ | Evaporation over inflow ratio [‰] of Groß Glienicke Lake in 2023 | 42.28 | 29.07 |



**Hydrological modeling and model evaluations**


Figure 8 illustrates the simulated vs. observed hydraulic heads (meters above sea level: m
a.s.l.) at West-GGS Piezometer (W-GGS), Lake Gross Glienicke (GGS), and East-GGS
Piezometer (E-GGS) over the period from January 2008 to December 2023. The model's
performance is evaluated using several metrics, including the Kling-Gupta Efficiency (KGE),
Percent Bias (PBIAS), and Root Mean Square Error (RMSE), as shown in Table 3. A strong
alignment is evident between simulated and observed hydraulic heads, both in terms of
magnitude and seasonality.
The simulated groundwater levels on the west side of the lakes, despite some over- and
underestimations, showed very good agreement with the observed data. For the calibration
period (2008-2018), the performance metrics are KGE of 0.86, PBIAS of 0.0%, and RMSE of
0.13 m. During the validation period (2019-2023), the model maintained high performance
with a KGE of 0.82, PBIAS of -0.1%, and RMSE of 0.07 m. Both observed and simulated
data exhibit a general declining trend over the study period, with hydraulic heads decreasing
from approximately 30.15 m to 30 m.
For GGS, similar to W-GGS, both observed and simulated values show a decreasing trend
from around 31 m in 2008 to approximately 29.80 m in 2023. The model simulations closely
follow the observed data, with minor deviations. The performance metrics for GGS during the
calibration period are KGE of 0.78, PBIAS of -0.2%, and RMSE of 0.13 m. In the validation
period, the metrics are KGE of 0.75, PBIAS of 0.0%, and RMSE of 0.06 m, indicating high
accuracy in representing the hydraulic behavior of the lake.
The observed and simulated groundwater dynamics on the east side of GGS show good
agreement. During the calibration period, the performance metrics are KGE of 0.84, PBIAS
of 0.1%, and RMSE of 0.09 m. In the validation period, the metrics are KGE of 0.70, PBIAS





of 0.2%, and RMSE of 0.09 m. Overall, the high-performance metrics confirm the model's
reliability and accuracy in capturing both the long-term trends and seasonal variations of
groundwater-surface water dynamics within the study area, providing valuable insights into
groundwater-surface water interactions.

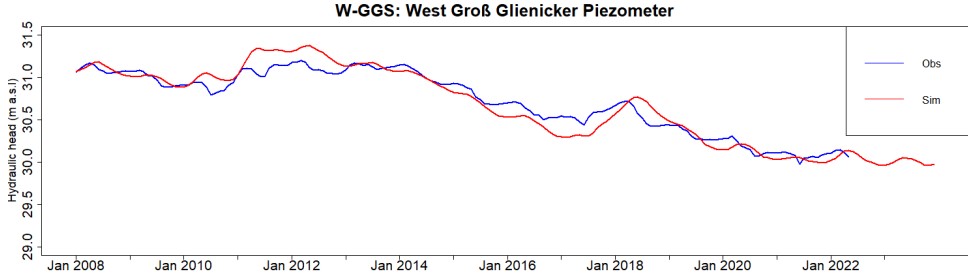

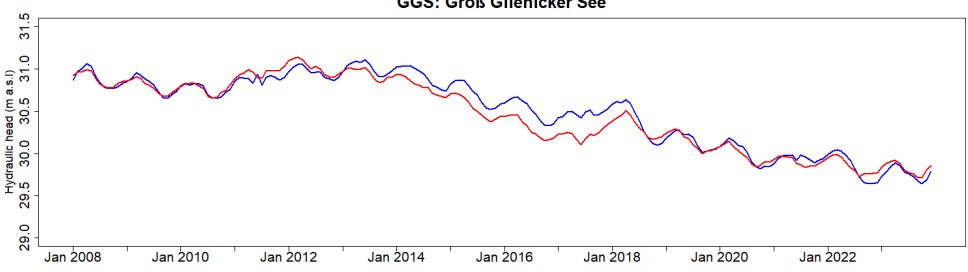

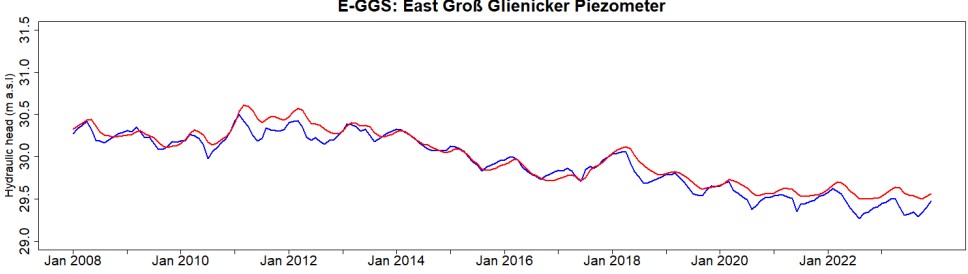


**Fig. 8.** Time series of observed and simulated hydraulic heads at three locations: (a) West Gross Glienicke
Piezometer (W-GGS), (b) Lake Gross Glienicke (GGS), and (c) East Gross Glienicke Piezometer (E-GGS) from
January 2008 to December 2023.
**Table 3.** Model performance evaluation using several metrics for both calibration (2008-2008) and validation
(2018-2023) periods

|  |  | West Gross Glienicke | East Gross Glienicke | Gross Glienicke Lake |
|---|---|---|---|---|
| **Calibration** | **KGE** | 0.86 | 0.84 | 0.78 |



| | | | | |
|---|---|---|---|---|
| **2008-2018** | **PBIAS** | 0.0 | 0.1 | -0.2 |
| | **RMSE** | 0.13 | 0.09 | 0.13 |
| **Validation 2019-2023** | **KGE** | 0.82 | 0.7 | 0.75 |
| | **PBIAS** | -0.1 | 0.2 | 0 |
| | **RMSE** | 0.07 | 0.09 | 0.06 |


The calculated annual water exchange for GGS from 2008 to 2023, detailing the inflow to the
lake, outflow from the lake, and their differences (net flow = inflow - outflow) are presented
in Figure 9. The inflow to GGS demonstrates substantial annual variability. For instance,
years such as 2008, 2011, 2012, and 2018 show relatively higher inflows compared to other
years. Notably, there is a discernible decreasing trend in inflow from 2011 to 2021, with 2018
being an exception. This trend indicates a progressive reduction in the hydrological inputs to
the lake over the decade. Simultaneously, the outflow from GGS shows significant annual
variability, with the highest outflow occurring between 2012 and 2017. This increased
outflow, coupled with the decreasing inflow, points to a period of significant net water loss
for GGS, potentially impacting the lake's water levels. Positive net flow values in years 2011,
2018, and 2022 indicate years when inflow exceeds outflow, contributing to the lake's water
gain. Variation in both water gain and loss over the study period, reflects the complex
interplay of natural hydrological processes governing the lake's water balance.
Figure 10 illustrates the water exchange dynamics of GGS over the period from August 2022
to September 2023, highlighting seasonal patterns in inflow, outflow, and net flow. During
the warmer months, particularly June and July, the inflow to the lake peaks at approximately
90,000 m³, indicating a significant increase in water input during this period. Among the
summer months (July to September), August has the lowest amount of inflow, with an
average of approximately 40,000 m³. In contrast, the inflow is pronouncedly lower during the
months of March, April, and May, averaging around 20,000 m³. The lake experiences positive
net flow, reflecting water gain, during the months of June and July. From December to May
(except February), the net flow is predominantly negative, indicating that outflow exceeds
inflow. During these months, the lake loses water, with outflow reaching its highest levels.





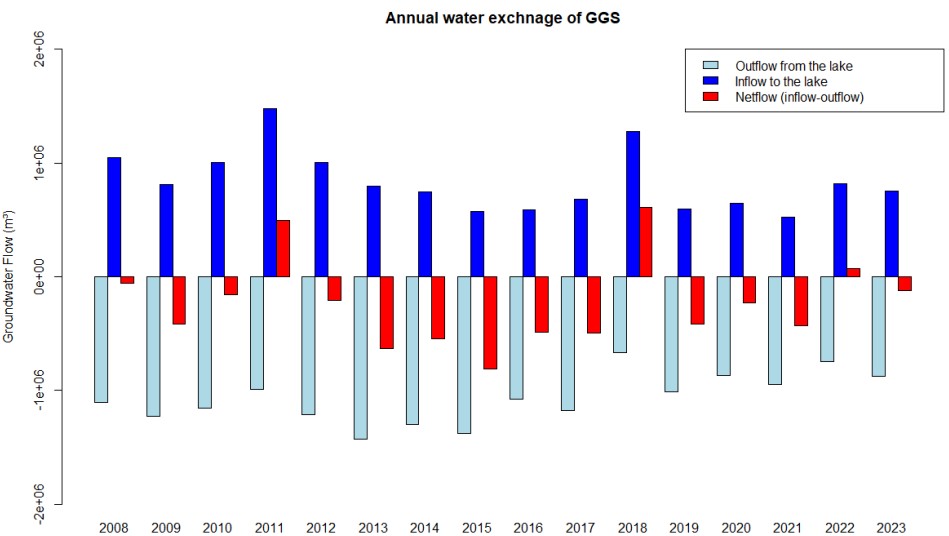


**Fig. 9.** Annual water exchange of Lake Groß Glienicke (GGS) from 2008 to 2023, showing the inflow to the lake, outflow from the lake, and net flow (Outflow - Inflow)

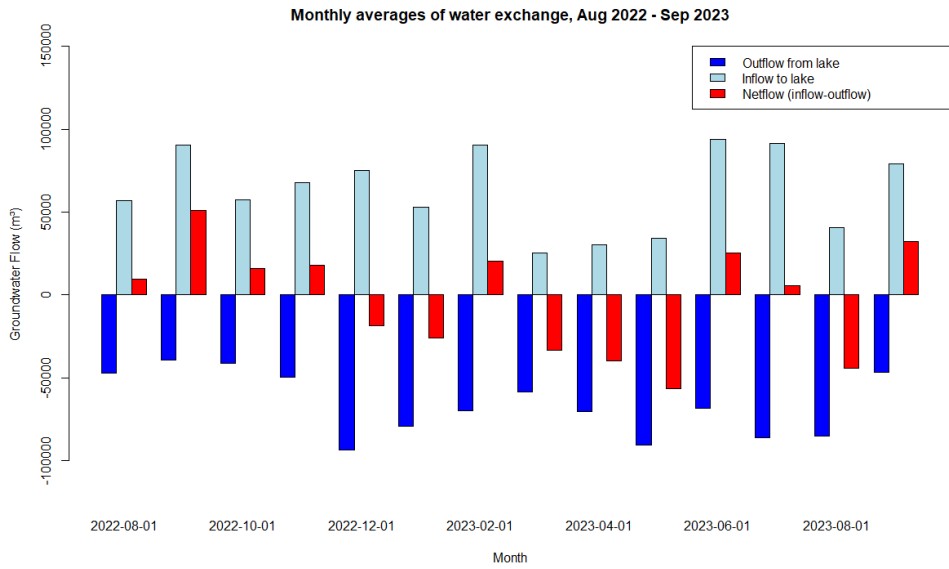


**Fig. 10.** Monthly water exchange of Lake Gross Glienicke (GGS) from August 2022 to September 2023




To compare the lake evaporation values calculated by the HGS model and the results of the
isotopic signature of lake water actual evaporation (E) from GGS for the years 2022 and 2023
was calculated using Eq. 11, which incorporates simulated annual water inflow to GGS ($I_{HGS}$)
from the HGS model and the E/I ratio derived from isotope analysis (mean ratio: 37%). These
results were compared to the evaporation rates from GGS simulated by the HGS model (Fig.
11). The annual evaporation estimates from the isotope analysis for the years 2022 and 2023
show good agreement. The values calculated by the HGS model ($E_{HGS}$) are slightly lower
(around 80 mm). The general agreement between the evaporation rates simulated by the HGS
model and the values derived from the isotope approach indicates accurate inflow simulation
by the HGS model. Transferring the *E/I* ratios from 2022 and 2023 to calculate evaporation
for earlier years (2015-2021, Fig. 11) results in notable discrepancies, particularly evident in
2015 and 2018, where evaporation reaches 450 and 1000 mm, respectively, compared to $E_{HGS}$
values of 590 and 614 mm for those years. However, modified isotope analysis (considering
annual variations in inflow and temperature, see Eq.11) demonstrates good agreement with
the HydroGeoSphere model, emphasizing our approach of incorporating temperature and
inflow data for accurate evaporation predictions for earlier years. This suggests that while the
*E/I* ratio obtained from 2022 and 2023 can be applied to estimate evaporation in previous
years, adjustments for differences in inflow and temperature between those years and 2022-
2023 are crucial for enhanced estimations of evaporations in years without isotope analysis
(E/I).

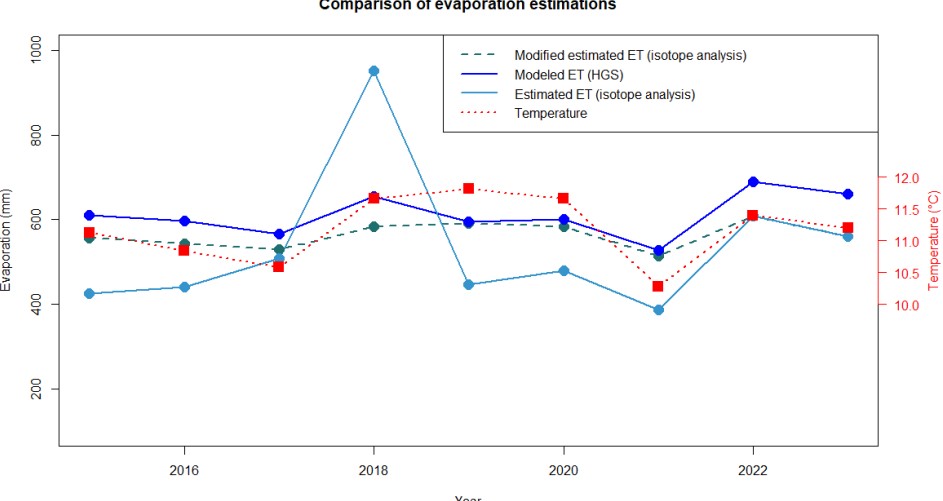

**Fig. 11.** Comparison of actual evaporation from Lake Gross Glienicke (GGS) estimated by Hydroeosphere model (HGE) and isotope analyses (E/I ratio), and modification of isotope analyses (E/I ratio) considering annual temperature and inflows to the lake. Annual temperatures are indicated, corresponding to the secondary y-axis.

## 4. Discussion

The comparison of actual evaporation (E) from GGS for 2022 and 2023, derived from isotope analysis (E/I) and HydroGeoSphere (HGS) model calculations, demonstrates the comparability of these methods. Despite slight differences in the results, likely due to the influence of direct precipitation with lighter isotope signatures diluting the lake water, the general consistency between the methods underscores the accurate simulation of inflows by the HGS model. Seasonal variations in water isotopic compositions offer a valuable perspective for evaluating hydrological model simulations. Higher outflow observed from December 2022 to May 2023 provides insights into the dilution phase of groundwater isotopic compositions on the eastern side of GGS during these months. Notably, GGS water isotopes reached their heaviest form in August 2023, contrasting with other summer months with similar temperatures and precipitation, suggesting lower inflow during that particular month as simulated by the HGS model. This approach highlights the value of using isotopic data to evaluate model simulations, providing a complementary angle to traditional methods. Considering the higher precipitation and warmer temperatures in the summer of 2022 compared to the summer of 2023, the isotopic composition of both lake water and east





groundwater was heavier during the summer of 2022. This indicates that isotopic "light"
rainwater, which directly recharged the lakes in the summer  of 2022, was insufficient to
counterbalance the influence of the higher temperatures, resulting in a heavier isotopic
composition in the lakes. Despite the high precipitation in summer 2022, which would
typically dilute the isotopic signature of the lake, the higher temperatures (evaporation) led to
an enrichment of water isotopes. This phenomenon can be attributed to the  large water body
of the lake (approximately 4 million m³ considering the average water depth of 6.5 m reported
by Wolter and Ripl (1999) compared to the rain amount (0.05 million m³), where the
relatively small volume of rainwater was not enough to significantly alter the isotopic
composition of the much larger lake volume. This is in line with the findings of the study by
Vyse et al.2020 where they discovered a more significant influence of rain on shallow water
bodies compared to larger water bodies in the State of Brandenburg (NE-Germany). The low
impacts of rainwater on the isotope composition of the lake can also be interpreted to mean
that GGS is a groundwater-dominated lake with a very good hydraulic connection to the
groundwater, which keeps the lake water fresh and diluted by providing a source of
isotopically-depleted water. This underlines the robustness of the isotopic approach against
the variations of meteorological factors influencing the isotopic signature.
The isotopic differences between lake water and groundwater on both side of the GGS,
highlight the complex interactions and distinct hydrological processes occurring within the
study area. Considering that the groundwater samples from the east and west of GGS belong
to the same depth (9-10 m below surface) and same aquifer, the isotopically-enriched
groundwater in the east side can only be explained by a well-mixed interaction of lake water
and groundwater in this area. This underlines that GGS is a flow-through lake, a conclusion
supported by the E/I ratio being up to 37% (e.i., mean ratio) in 2022 and 2023. The E/I ratio
of GGS aligns with the E/I ratio reported for wetlands and lake water bodies downstream of
the Spree catchment showing similar climate conditions. In 2021, the E/I ratio in this area was
up to 34% (Chen et 625 al., 2023). Vyse et al. (2020) reported that the wetlands with lower
landscape elevations located in northen Brandenburg typically possessed higher E/I ratios
than the ones with higher elevations. This is due to the hydrological function of small
waterbodies in the Pleistocene  landscape. Higher wetlands have a more recharge/flow
through character whereas lower positions show a discharge character. Moreover, Cluett and
Thomas (2020) highlighted that the sensitivity of lake water isotopes to inflow and
evaporation can vary significantly over time, influenced by regional hydroclimate (e.g., direct
precipitation and humidity) and local  hydrology (e.g., type of the lake). The uncertainty in





evaporative loss calculated using the code embedded in the Hydrocalculator is mainly due to
uncertainties in the required inputs, temperature, and humidity, which can cause variations of
up to 2% (E/I) according to Skrzypek  et al. (2015). Despite these potential measurement
variations, the calculated E/I ratio for GGS  provides a reliable estimate that aligns with
known hydrological behaviors in similar regions.
For the period 2015-2021, evaporation estimates derived from the isotope analysis (E/I) of
2022 and 2023 generally show lower values compared to the HGS model's evaporation
estimates, except 2018. This year saw a strong inflow peaking due to heavy rainfall events in
the summer 2017. This suggests that using the E/I ratios from 2022 and 2023 for earlier years
without  adjustments can lead to significant inaccuracies. The modified isotope analysis (E/I),
which incorporates annual variations in inflow and temperature, shows a better agreement
with the  HGS model evaporation calculates, especially in the earlier years. This finding
underscores the  importance of including both temperature and inflow data for more accurate
evaporation  predictions. Comparing these results with previous studies, Herbst and Kappen
(1999) reported evaporation from the Bornhöved Lake, which covers 1.1 km² in northern
Germany and has a  maximum depth of 26 m, to be around 650 mm for the years 1992-1995.
The evaporation estimates for GGS from both the HGS model and the modified isotope
analysis fall within a  similar range reported by Herbst and Kappen (1999), suggesting that
despite differences in  geographical and hydrological characteristics, the annual evaporation
rates for German lowland  lakes are comparable. These findings support the use of
comprehensive, multi-faceted  approaches in hydrological studies to improve the precision of
evaporation estimates and  enhance water resource management.

**5.  Conclusion**
This study has addressed the challenge of accurate estimation of water balance components
comprehensively  through  the  quantification  of  subsurface-groundwater  inflow  and
evaporation losses to Gross Glienicke Lake (GG), located in northeast Germany. Through the
combined use of the isotopic mass balance model, HydroCalculator, alongside the fully
integrated hydrological model, HydroGeoSphere (HGS), a detailed understanding of the
hydrological dynamics governing GG Lake was attained. The calculated evaporation rates
derived from the isotopic mass balance model, exhibit strong alignment with the actual
evaporation rates calculated by the HGS model. This alignment underscores on one hand the
reliability and efficacy of the integrated hydrologic modeling approach in predictions of water





balance components such as inflow to the lake in a complex hydrogeological setting. On the
other hand, incorporating evaporation rate estimations given by isotope analysis corrected by
temperature variations and historical inflows leads to an improvement of the inflow results
even for the years without measured isotope data. Despite inherent uncertainties associated
with water isotope signature analyses, the integration of isotopic data with hydrological
modeling has provided valuable validation for the estimation of water balance components.
Moving forward, this integrated approach holds promise for enhancing the robustness of
hydrological models and facilitating more accurate assessments of water resources and
ecosystem dynamics in similar lake environments.

**Data availability:** All data (except the data provided by Berliner Wasserbetriebe) used to
process and set up the models will be available upon request. Data provided by Berliner
Wasserbetriebe can be requested through a separate usage agreement with Berliner
Wasserbetriebe.
**Author contributions:** Data collection, fieldwork, HGS model setup, and code development,
model input-output analysis, writing (original draft, review, and editing). JS: Isotopic
laboratory analysis, writing (review and editing). MS: Model input-output analysis, writing
(original draft, review, and editing). CM: Model input-output analysis, writing (original draft,
review, and editing).
**Competing interests:** The authors declare that they have no conflict of interest.
**Acknowledgments:** This investigation was funded through the Einstein Research Unit
'Climate and Water under Change' from the Einstein Foundation Berlin and Berlin University
Alliance (ERU-2020-609). We would like to thank Reinhard Hinkelmann, Franziska Tügel,
and Can Ölmez from the Technical University of Berlin for providing potential evaporation
data. We like to thank Dr. Gunnar Lorenzen and Bertram Monninkhoff (Berliner
Wasserbetriebe) for providing us with the FEFLOW model as well as the actual data on the
drinking water extraction rates around the Havel River. We are grateful to our colleague,
Dieter Scherer his rainwater sampling assistance, and Patrick Zentel for his field and
laboratory assistance.

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
