# Peer review of "compositions with a hydrological model"

_Hydrology and Earth System Sciences, 2024_

## Referee Comment (RC2)

[referee-annotated manuscript omitted]

---

## Author Comment (AC1)

**Reply to Referee #1**

Dear Referee,

We are grateful for your insightful comments on our manuscript. Your feedback has contributed to enhancing the quality of our work. Below, we provide a point-by-point response (in blue) to your comments (in black) and outline how we will address each suggestion in the revised manuscript.

Sincerely,

Nariman Mahmoodi, Ulrich Struck, Michael Schneider, and Christoph Merz
* * *
The manuscript titled "Reinforce Lake Water Balance Component Estimations by Integrating Water Isotope Compositions with a Hydrological Model" has been thoroughly reviewed. It presents an interesting study with valuable practical applications. However, the reviewer has noted the following concerns for the authors' and editor's consideration:

1. In the abstract (lines 13-14), the authors suggest their approach as an alternative method for capturing the dynamic behaviour of the hydrological groundwater/surface water system, yet the study is based on only one year of sampling. Can this work truly represent the hydrological dynamics of the lake system? Additional clarification or rephrasing may be needed.

   We agree that relying on only one year of isotope analysis may not be sufficient to claim that this study fully represents the long-term hydrological dynamics of the lake system. Our intention in the original statement was to emphasize that the approach we used—integrating isotope analysis with hydrological modeling—offers an alternative to previous studies that focused solely on model outputs without incorporating direct measurements of isotope data. We acknowledge that long-term monitoring would provide a more robust representation of the system's dynamics. We will rephrase the abstract to clarify that this approach is intended as a complement to more extensive modeling efforts. The revised wording will better reflect the scope and limitations of the study to avoid misunderstandings.

2. The authors state that an isotope mass-balance model was used to quantify the evapotranspiration rate by accounting for groundwater inflow to offset evaporation losses, in the context of the lake's water balance. However, how is open water evaporation handled? Does the evapotranspiration calculated in this study include ET from groundwater? Further clarification on this point would be beneficial.

   In our model, we employed an isotope mass balance approach, which compares the isotopic composition of water between groundwater and lake

water. The changes in isotope composition of lake water in comparison to the groundwater will be used to calculate the evaporation form the lake not evapotranspiration. More details of how the isotope mass balance works can be seen in Skrzypek et al. (2015). We will modify the manuscript to make that clear.

The referee has inquired about whether we accounted for direct evaporation from groundwater. Given that our mass balance model focuses on comparing isotope compositions, the specific pathway of evaporation—whether it occurs directly from groundwater or elsewhere—is not critical to the model's primary function. However, it is important to note that groundwater can influence soil moisture in the root zone and surface evaporation when the water table is near the surface, such as in wetland areas. When the groundwater table is within or close to the model's soil column, it can substantially affect soil moisture levels and evapotranspiration rate as a consequence. For further reference, see Chen and Hu (2004). In our study area, the GGS catchment, water flow in the root zone is predominantly vertical, moving downward to the groundwater table which is not connected to the root zone. The one-dimensional vertical flow, along with processes like evaporation and transpiration, has been accounted for in our hydrological model.

**References:**

Skrzypek, G., Mydłowski, A., Dogramaci, S., Hedley, P., Gibson, J.J. and Grierson, P.F., 2015. Estimation of evaporative loss based on the stable isotope composition of water using Hydrocalculator. Journal of Hydrology, 523, pp.781-789.

Chen, X. and Hu, Q., 2004. Groundwater influences on soil moisture and surface evaporation. Journal of Hydrology, 297(1-4), pp.285-300.

3. The authors mention a hydraulic connection between the lake and groundwater system. Additional details on the assumptions made would be valuable. For instance, is there any seepage from the lakebed to the groundwater?

Based on lake level records and groundwater data monitored in piezometers, the groundwater fluctuations on both sides of the lake closely mirror the lake level fluctuations (Fig. 1). This indicates a strong hydraulic connection between the lake and the surrounding groundwater, supporting the conclusion that the lake is a flow-through system, as also confirmed by the isotope composition analysis, where E/I ratio is around 40% which is typical for flowthrough systems. Additionally, there is no surface water inflow to the lake. To clarify, determining the precise nature of the connection between the lake and groundwater through the lakebed would require specialized sediment sampling and laboratory analysis, which falls outside the scope of this study. In opposite, the isotopic composition analysis includes the horizontal and vertical exchange between

groundwater and lake water body. The inflow to the lake has been calculated for the whole lake domain (including lakebed influence) and therefore can be used to validate the results of the hydraulic modeling.

4. The authors used the HydroGeoSphere (HGS) modelling code (Aquanty Inc., 2023) to simulate hydrological processes in the study area. Could the authors clarify why the HGS model was selected over other 3D models, such as MODFLOW, and discuss any comparative advantages?

We selected HydroGeoSphere (HGS) for this study because it provides a fully integrated simulation of both surface and subsurface hydrological processes, which was crucial for capturing the dynamics of the study area. Unlike MODFLOW, which is primarily a groundwater flow model, HGS offers a comprehensive approach by coupling surface water, groundwater, and soil moisture interactions. This allows us to simulate processes such as evaporation from the lake (or other land use- land covers), overland flow (if any), and the interaction between surface water and groundwater, all of which are key to understanding the hydrological balance in our study area. Another advantage of HGS is its ability to model unsaturated and saturated flow, which is a basic prerequisiteto calculate the groundwater recharge directly. Additionally, HGS's ability to simulate evaporation from the lake—an essential process in this case—would be difficult to achieve with MODFLOW without significant customization or the use of third-party extensions. Therefore, the use of HGS provided a more comprehensive approach to simulating the entire hydrological system, allowing us to capture the multi-dimensional water flow pathways and interactions with greater accuracy. This uniqueness of the HGS model will be added to the manuscript.

5. It would be helpful to provide more information about the HGS model setup, including the number of aquifer layers, initial boundary conditions, and model parameters used in this study.

The HydroGeoSphere (HGS) model used in this study integrates both surface and subsurface flow components to simulate groundwater-surface water interactions. The model setup includes multiple subsurface layers of aquifers and aquitards, the characteristics of which have already been detailed in the manuscript. The hydraulic properties and parameters of these layers will be provided in the supplementary material.

The initial conditions for subsurface and surface flows were established by running the HGS model under steady-state conditions. Predefined groundwater and surface heads were used as the starting point for the transient simulation. Lateral boundaries were defined with specific node sets along the Havel River (constant head boundary), representing flow exchange across these boundaries.

Different land use and land cover types (forest, grassland, urban, and agriculture) were assigned distinct properties for overland flow simulations, including obstruction storage height, rill storage height, and coupling length. Evapotranspiration (ET) parameters—such as leaf area index (LAI), root depth, and evaporation depth—were specified for each land cover type. In the HGS model, ET combines plant transpiration and evaporation, affecting both surface and subsurface flows. Plant transpiration within the root zone depends on LAI, nodal moisture content ($\theta$), and a root distribution function (RDF) applied to a defined extinction depth. Depth-dependant evaporation is modeled using quadratic depth decay function.

LAI data was measured for different land types and compared to the MODIS dataset to provide time-varying LAI inputs for the model. In the HGS model, potential evapotranspiration (PET) is set as a boundary condition, representing the (highest) amount of water that would evaporate and transpire if the water table were at the surface. PET was calculated based on energy balance methods, particularly applied to Lake GGS.

This setup allows for a detailed simulation of surface and subsurface processes, capturing complex interactions such as lateral flow, groundwater abstraction, evapotranspiration, and overland flow. Further explanations and details on the model setup and parameters will be included in the manuscript, with additional information provided in the supplementary material.

General comment: The manuscript is engaging, though minor grammatical and punctuation errors are present. Addressing these would improve clarity and readability.

Thank you for bringing this to our attention. To ensure clarity, minor grammatical errors, typos, and sentence rephrasing will be addressed in the revised manuscript.

---

## Author Comment (AC2)

**Reply to Referee #2**

Dear Referee,

We sincerely appreciate your time and thoughtful review of our manuscript. Your insightful comments and careful revision have improved the quality of our work and helped us present our results accurately. Below, we provide a point-by-point response (in blue) to your main comments (in black) and outline how we address each suggestion in the revised manuscript.

Sincerely,

Nariman Mahmoodi, Ulrich Struck, Michael Schneider, and Christoph Merz
* * *
1. The authors combine a hydrological modeling approach and an isotopic modeling approach to understanding lake water balance issues within a lake in Germany that has been declining in lake water levels. While I think the approach is interesting and has potential, I have serious concerns about the isotopic modeling as presented. While the isotopic theory they present is mostly correct, and they are using the Hydrocalculator developed by Skrzypek et al 2015, the parameters they present in Table 2 as input to the Hydrocalculator are so far out of line, I'm surprised they actually got any numbers out that make any sense at all. I fear they may have calculated some critical values using ‰ values, when they should have been using absolute values (not multiplied by 1000). The literature can be confusing on this point, but using the wrong value would lead to the negative and completely out of bounds values they give for kinetic fractionation and other values. I give detailed comments on the pdf directly, but these errors on input for the hydrocalculator might explain why the authors had to "adjust" their isotopic estimates of evaporation before coming to something that might be reasonable. I am also unclear on many steps that they take in their methods. Until they revisit these data, I can't really evaluate the rest of the manuscript. The isotope data themselves look OK, and I am hoping they can redo the analysis without too much problem, once they find the error that led to those values in table 2.

   We appreciate the reviewer's careful assessment of our work and the constructive feedback regarding the isotopic modeling approach. We acknowledge the concerns raised about the parameters presented in Table 2 and the potential errors in the input values used in the Hydrocalculator.

   Upon reviewing our submission, we discovered that an incorrect version of Table 2 was inadvertently included in the manuscript during the final stage of preparation before submission. This mistake resulted from inadvertently carrying over the table from an earlier draft that contained outdated parameter

values. We deeply regret this oversight and understand that it significantly impacted the reviewer's ability to evaluate our isotopic modeling approach.

To address this issue, we replaced Table 2 with the correct version ( presented below as Table 2) and an additional table (presented below as Table 1) containing the input values. We have also carefully rechecked our calculations to ensure that all output parameters, including fractionation factors, were applied correctly following the guidelines in Skrzypek et al. (2015). Additionally, we have revised the Methods section to clarify our step-by-step approach in the isotopic modeling, ensuring transparency and reproducibility.

The reviewer's comments have been invaluable in helping us identify and rectify this error. We hope that with these corrections, our analysis is now clear, and we look forward to any further suggestions.

In addition, to clarify the integration of the isotope analyses and hydrological models, we provide a more detailed explanation here:

To validate the accuracy of our hydrological model, we tested how well it could estimate lake evaporation. We compared its results with evaporation values derived from isotope analysis (HydroCalculator), an independent method. If both approaches produced similar results, it confirmed that the model accurately simulates water fluxes, including lake inflow and evaporation.

Since isotope data was unavailable for earlier years (2015–2021), but significant changes in lake levels were observed during this period, we extended our analysis to estimate evaporation for these years. However, the evaporation-to-inflow (E/I) ratio from recent years (2022–2023) could not be directly applied to earlier years due to variations in temperature and inflow, which influence isotopic signatures through dilution and enrichment.

To account for these variations, we incorporated annual temperature and inflow differences into our model. Specifically, we calculated temperature and inflow ratios by comparing each year (Yx) to the reference period (2022–2023). These ratios were then used to adjust the E/I values for earlier years. For example, if a given year was warmer than 2023, it would have experienced higher evaporation and isotopic enrichment, requiring an increased E/I ratio. Similarly, if inflow was lower in a particular year, evaporation effects would be more pronounced, further influencing the E/I adjustment.

The adjusted E/I ratios were then applied to refine evaporation estimates for 2015–2021, which were subsequently compared to the evaporation rates simulated by the hydrological model. The strong agreement between these estimates enhances confidence in the model's ability to accurately simulate key water balance components over extended periods.

This approach allows us to reconstruct evaporation dynamics in years without direct isotope measurements, improving our understanding of historical lake and groundwater variations.

**Table 1.** Input Data for the Hydrocalculator Model

|  | $\delta^2H$ | $\delta^{18}O$ | Climate data |
|---|---|---|---|
| Pool start (lake winter sample) | -16.160 | 0.090 |  |
| Pool final (lake summer sample) | -10.660 | 1.370 |  |
| Precipitation | -65.715 | -11.790 |  |
| Mean temperature (˚c) |  |  | 20 |
| Relative humidity (%) |  |  | 60 |

**Table 2.** Variables used for the calculation of evaporative losses

| Parameters | Description | $\delta D$ | $\delta^{18}O$ |
|---|---|---|---|
| $\varepsilon_k$ | Kinetic isotope fractionation factor [‰] (h dependent) | 5 | 5.68 |
| $\varepsilon^*$ | Equilibrium isotope fractionation factor [‰] (T dependent) | 84.355 | 9.778 |
| $\varepsilon$ | Total isotope fractionation [‰] | 82.793 | 15.363 |
| Ck | Kinetic isotope fractionation constant [‰] | 12.5 | 14.2 |
| $\alpha^*$ | Equilibrium isotope fractionation factor [‰] (T dependent) | 1.0844 | 1.0098 |
| $\delta^*$ | Limiting isotope composition | -0.4727 | 4.358 |
| m | Enrichment slope | 1.277 | 1.441 |
| $\delta_A$ | Ambient air moisture | -138 | -21 |
| $E/I_{Y2022}$ | Evaporation over inflow ratio [%] of Groß Glienicke Lake in 2022 | 43.37 | 29.63 |
| $E/I_{Y2023}$ | Evaporation over inflow ratio [%] of Groß Glienicke Lake in 2023 | 42.28 | 29.07 |

How were the water samples collected?  Was the water within the well pumped purged before sampling?

The following procedure was followed for water sampling:

Before sampling, the total water depth was measured to calculate the volume of stagnant water inside the piezometer. At least three well volumes of water were then pumped. Pumping continued until field parameters (temperature, electrical conductivity, and pH) stabilized, ensuring that the sample was representative of the aquifer rather than stagnant casing water. After sampling, the collected water was stored in a cooling box and transferred to the cooling room and laboratory. This procedure will be added to the manuscript.

2. Likely VSMOW, VSLAP and GISP were not analyzed in the same set as your samples.  What standards, and how many were analyzed with your set, and did you have an independent standard (not used in the calibration regression) to calculate accuracy?  Did you take field duplicates to calculate precision?

Laboratory analyses were performed by using VSMOW Gisp and VSLAP at the start of each sample run. For example, on a daily basis, we used the sample vials of the three standards to confirm the calibration which is performed once

a while (about every half a year) on fresh samples of the three calibration standards. Over the past 15 years, the internal calibration of the Picarro instrument has not changed significantly.

3. Usually, the first couple of injections are deleted, unless all samples are very close in range as the injections have significant carryover from the previous sample.  Was this evaluated?

No, the carry-over effect of the system is highly dependent on the isotopic difference between samples. In the case of a $\delta^{18}O$ difference of 1 ‰, no significant carry-over effect is observed. For differences greater than 5 ‰ in $\delta^{18}O$, the carry-over effect influences only the first two injections by approximately 0.5 ‰, depending on the isotope composition of the prior sample. The remaining four injections stay unaffected and show no trend.

What was an acceptable sigma between injections? If you actually did run VSMOW and GISP, this carryover effect can be huge.

Exactly, find here an uncorrected actual set of Standard Samples including six injections as a startup in a sample sequence. Since "Gisp" is sold out we got our own standard sample "TapwaterSTD".

**Table 2.** Raw data for the first three standard samples, each measured six times in replicate (output of Picaro instrument). The "ignore" and "good" columns are generated as a helpful guide to indicate which injections are considered good or bad (to be ignored).

| Line | Analysis | Raw $\delta^{18}O$ Measurements | $\delta^{2}H$ Mean | $H_2O$ Mean | Ignore | Good |
|------|----------|-------------------------------|-----------|-----------|--------|------|
| Slap | P-5274 | -48,766 | -350,391 | 16882 | -1 | 0 |
| Slap | P-5274 | -53,131 | -406,01 | 17327 | -1 | 1 |
| Slap | P-5274 | -54,1 | -420,473 | 17446 | -1 | 1 |
| Slap | P-5274 | -54,805 | -427,223 | 17542 | 0 | 1 |
| Slap | P-5274 | -54,775 | -429,082 | 17608 | 0 | 1 |
| Slap | P-5274 | -54,657 | -430,547 | 17722 | 0 | 1 |
| TapwaterSTD | P-5275 | -14,19 | -136,45 | 17768 | -1 | 1 |
| TapwaterSTD | P-5275 | -9,492 | -84,021 | 17990 | -1 | 1 |
| TapwaterSTD | P-5275 | -7,994 | -68,112 | 17978 | -1 | 1 |
| TapwaterSTD | P-5275 | -7,872 | -63,604 | 17873 | 0 | 1 |
| TapwaterSTD | P-5275 | -7,871 | -59,801 | 18042 | 0 | 1 |
| TapwaterSTD | P-5275 | -7,816 | -60,205 | 17907 | 0 | 1 |
| SMOW | P-5276 | -0,847 | -16,212 | 17949 | -1 | 1 |
| SMOW | P-5276 | -0,564 | -9,799 | 17958 | -1 | 1 |
| SMOW | P-5276 | 0,036 | -5,905 | 17888 | -1 | 1 |
| SMOW | P-5276 | 0,182 | -4,353 | 18000 | 0 | 1 |
| SMOW | P-5276 | -0,317 | -5,749 | 17923 | 0 | 1 |
| SMOW | P-5276 | 0,102 | -5,666 | 17944 | 0 | 1 |

4. Minor comments on the manuscript:

   To ensure clarity, all minor comments on typos and rephrasing will be addressed in the revised manuscript.

---

## Author Response (AR2)

**Reply to Referee #2**
* * *
This is the second time I have reviewed this manuscript, and I find the authors did a very good job in addressing my comments and concerns. I did find some minor typos, and incorrect equation references which I highlighted on the pdf. I think the paper nicely illustrates how modeling approaches and isotopic analysis can complement water balance calculations for lakes. Nice job.

Thank you very much for your thoughtful and supportive review. We appreciate you taking the time to review our manuscript again. As suggested, we have carefully revised the manuscript to correct minor grammatical errors and typos and have improved sentence clarity where needed.